# Analysis and Prediction of Flow-Induced Vibration of Convection Pipe for 200 t/h D Type Gas Boiler

**Shouguang Yao** [1,*] **, Xinyu Huang** [1]**, Linglong Zhang** [1]**, Huiyi Mao** [1] **and Xiaofei Sun** [2]

1   School of Energy and Power Engineering, Jiangsu University of Science and Technology,
    Zhenjiang 212100, China; 17719491172@163.com (X.H.); 4188645193@163.com (L.Z.);
    m19895833055@163.com (H.M.)
2   Key Laboratory of Energy Thermal Conversion and Control of Ministry of Education,
    School of Energy and Environment, Southeast University, Nanjing 210096, China; ntsunxf@126.com
*   Correspondence: zjyaosg@126.com; Tel.: +86-15051110000

**Abstract:** This paper is aimed at the analysis and prediction of the fluid-induced vibration phenomenon in the convection tube bundle area caused by Karman vortex street shedding in the background of a 200 t/h large-capacity D-type gas boiler. Based on the numerical simulation of flue heat state flow field and fast Fourier transform, the lift coefficient curve of different monitoring areas and the corresponding Karman vortex street shedding frequency are obtained. The accuracy of the analysis model is validated by comparing Karman vortex shedding frequency with acoustic equipment standing wave frequency. In order to meet the design requirements of the 200 t/h D-type gas boiler for reliable and stable operation, the vibration characteristics and variation rules of a convection tube bundle in a D-type boiler under different working conditions are predicted.

**Keywords:** D-type gas boiler; numerical simulation; Karman vortex shedding; flow-induced vibration

## 1. Introduction

With the development of boilers towards the direction of large capacity and high parameters, the causes of vibration of boiler equipment are more and more complex [1,2]. Fluid-induced vibration has become a hot issue in the boiler industry [3]. The vibration of the heat exchange tube in the convection tube bank is a fluid-induced vibration [4] which is mainly caused by the similarity between the periodic Karmen vortex shedding frequency in a flow tube bundle and the equipment's acoustic standing wave frequency. Many scholars have conducted much study work about the Karmen vortex which forms periodic shedding behind the tube bundle and causes the fluid-induced vibration when the fluid flows around the tube bundle [5–7]. Min [8] and Lai et al. [9] simulated the generation of the Karmen vortex and vortex shedding around the tube handle by computational fluid dynamics and used computational fluid dynamics software to carry out a numerical simulation of the generation of the flow around the tube bundle and the process of it evolving through vortex shedding. They calculated and analyzed the Karmen vortex vibration induced by vortex shedding. By the failure fracture phenomena of the HE302 Titanium condenser pipe bundle based on the simulation of the flow around a two-dimensional unsteady elastic support cylinder, Wang et al. [10] proposed and verified the feasibility of the fluid–solid coupling method in the failure analysis of the heat exchanger. Finally, the stress distribution and failure reason of heat exchanger tube bundles were obtained. In light of the relatively severe vibration phenomenon existing in the steam air heater of some boilers, Du et al. [11] numerically simulated the flow field in the process of the flue gas flowing through the heater by FLUENT and drew the conclusion that slue gas's Karmen vortex shedding frequency and acoustic standing wave frequency coupling caused the vibration of the equipment. By the numerical simulation of flow-induced vibration of the bicylinder in transverse flows, Chen et al. [12] drew the conclusion that the interaction between fluid and structure in

different dynamic responses has an obvious impact on the size of the separation vortex. Meanwhile, the effect of flow-induced vibration is more obvious with a larger mass ratio. In terms of the flow-induced vibration issue of the bicylinder, S. Kim et al. [13,14] performed an experimental study on the suppression of flow-induced vibration by the elastic support cylinder model. The experiment result shows that the transverse pitch of the cylinder and the velocity of the incoming flow have an obvious influence on the fluid-induced vibration. The vibration can be effectively suppressed by adding a flexible polyethylene sheet on the surface of the cylinder. They also found that the optimal length and the minimum width of the flexible polyethylene sheet are 2 D–2.5 D and 0.7 L, respectively. It is also effective to suppress the flow-induced vibration by installing the sheet on the dorsal stagnation point of the cylinder. R.D. Blevins [15] developed a mathematical model of nonlinear cylindrical vortex-induced oscillation based on the experiment which is carried out by changing the amplitude and the flow velocity. The result shows that this model can be used to analyze and predict resonance, off-resonance, and time history vortex-induced oscillation. Aimed at avoiding the acoustic vibration of the tail flue of large power plants' boilers, based on the summary of theoretical analysis and a large amount of experimental data, Wei [16] systematically discussed the mechanism of the acoustic vibration of boilers' tail flue and proposed his opinions on how to control the acoustic vibration of boilers' tail flue. Aiming at solving the problem of fluid-induced vibration of boiler equipment in practical engineering, Chai et al. [17] and Hong et al. [18] performed calculations and analysis according to traditional empirical formulas and discovered that installing the shockproof clapboard can effectively improve equipment's acoustic standing wave frequency so as to avoid the resonance caused by the approximation between the Karmen vortex shedding frequency and acoustic standing wave frequency. From the existing literature reports, analyses of the fluid-induced vibration of practical engineering equipment using experiments and numerical simulations are mainly focused on small heat exchangers [19]. For large equipment, such as boilers, the calculation and analysis depend on traditional empirical formulas. There are fewer numerical analyses [20–23].

In this paper, the numerical simulation method is used to monitor the time history curve of the lift coefficient in different regions of the convection tube bundle area in a 200 t/h large-capacity D-type gas-fired boiler. The precise Karmen vortex shedding frequency is obtained by fast Fourier transform in the convection tube bundle area of D-type boilers under different operation situations. Further, by comparing the acoustic standing wave frequency of the equipment, the regularity of the D-type boilers' fluid-induced vibration under different operation circumstances is predicted to provide a theoretical foundation for the design, operation, and anti-vibration solutions of this type of boilers subsequently.

## 2. Numerical Calculation Models and Methods

### 2.1. Physical Model

Considering that the goal of this paper is to conduct a simulation analysis of the Karmen vortex shedding phenomenon occurring in the convection tube bundle area, the modeling analysis is only on the flow and heat transfer in the flue flow area from the outlet of the condenser tube to the extension section of the convection zone's outlet. The following assumptions [19] are made:

(1) Disregard the influence of gravity and the pipe flow field on flue gas flow.
(2) The gap between the baffle plate and the shell wall as well as the heat exchange tube is considered to be closely connected.
(3) Due to the large calculation area of the target, the influence of the outer membrane water wall tube wall on the flow area is ignored, and the wall surface is simplified into a non-slip plane.
(4) The inlet of flow field calculation in the flue is set at the inlet of the flue.

The physical model is established as shown in Figure 1. It monitors the flow field of five sections intercepted from top to bottom in the rectilinear direction at the bottom of the lower baffle plate in the convection tube bundle area where the Karmen vortex shedding tends to occur.

The Karman vortex street frequency and standing wave frequency formed by the flue gas flowing through the convective region are shown in Table 1.

**Table 1.** Shedding frequency and first-order standing wave frequency at rated operating conditions.

| Horizontal Pitch (mm) | Longitudinal Pitch (mm) | Convection Flue Width B/m | Outer Diameter d/mm | Flue Gas Velocity V/m s$^{-1}$ | Karman Vortex Street Frequency (Hz) | First-Order Standing Wave Frequency (Hz) |
|---|---|---|---|---|---|---|
| 116.5 | 121 | 6.7 | 57 | 12.4 | 56.56 | 49 |

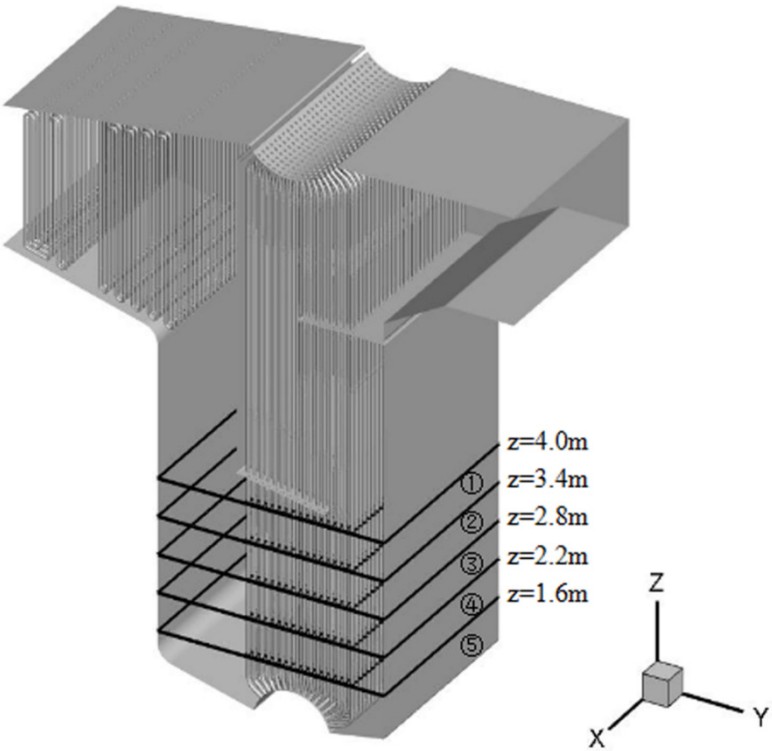

**Figure 1.** Schematic diagram of the lower end of the convection zone.

*2.2. Physical Mode*

In order to simulate the Karmen vortex shedding phenomenon in the convection tube bundle area of the boiler correctly, the thermal flow field of the flue was calculated according to the continuity equation, momentum conservation equation, energy conservation equation, and component transport equation.

The shedding frequency of Karman vortex shedding is:

$$f_p = S_{tr}\frac{V}{d} \tag{1}$$

where $V$ is the flue gas velocity; the unit is m/s; $d$ is Bundle diameter, the unit is m; $S_{tr}$ is the Strouhal number.

The turbulence model chooses the RNG $K$-$\varepsilon$ model. Considering that the radiation of gas-fired boilers is mainly caused by $CO_2$ and $H_2O$, the simulation chooses the discrete ordinates radiation (DO) model.

Since there is no burning in the gas flue of the boiler's convection tube bundle area and the diffusion and transport of different components exist, thus, in the thermal simulation, the chemical reactions are neglected in the diffusion of chemical components which adopts the mixed component hybrid calculation model. Namely, only the mixing process of calculation components is considered without the occurrence of chemical reactions.

On the premise of obtaining the thermal flow field of the convection tube bundle area, the following formulas are adopted to calculate the time history of the drag and lift round the tube integrals.

$$F_D^n = \oint (P_s^n \cos\theta + \tau_s^n \sin\theta) ds \tag{2}$$

$$F_L^n = \oint (P_s^n \sin\theta + \tau_s^n \cos\theta) ds \tag{3}$$

In the formula, the superscript $n$ means the value when $t_n = n\Delta t$; $F_L^n$ is the lift on the tube at the time $t_n$, $F_D^n$ is the drag on the tube at the time $t_n$; $P_s^n$ is the fluid pressure on the surface of the tube at the time $t_n$; $\tau_s^n$ is the shearing stress on the tube surface at the time $t_n$. Make the discrete solution of the above formula to obtain the lift curve of the Karmen vortex. Further, the Karmen vortex shedding frequency curve can be obtained through a fast Fourier transform (FFT).

### 2.3. Physical Parameters and Boundary Conditions

During the flowing heat transfer process of gas in the flue, its physical parameters will have a certain impact on the flow of heat transfer with the temperature changing. So, the physical parameters are density, specific heat capacity, thermal conductivity, viscosity, mass diffusivity, adsorption coefficient, scattering coefficient, and scattering phase function of the gas setting in the calculation.

The gas outlet is handled as the velocity inlet and the pressure outlet. The tube surface and the wall surface both adopt fixed wall temperature no-slip boundary conditions. The boundary conditions are summarized in Table 1.

Inside the gas flue of the boiler, there are bird nests on the superheater tubes, convection tube bundle, and water wall surface. Therefore, the surface emissivity that is related to the radiation heat transfer is the emissivity of the bird nest, while the emissivity of the bird nest is related to its status parameters. Correct experimental data are not easy to obtain. By looking up the emissivity value of the bird nest under different temperatures, the quadratic polynomial curve of the emissivity changing with temperature is obtained by fitting. At last, the thermodynamic parameters of wall surface boundaries are obtained based on thermodynamic calculations, as shown in Table 2.

**Table 2.** Summaries of inlet and outlet boundary conditions.

|  | Velocity/ms$^{-1}$ | Pressure/Pa | Temperature/K | Hydraulic Diameter/m |
|---|---|---|---|---|
| Gas inlet | 20.3 | −18.5 | 1473 | 2.795 |
| Gas outlet | — | −276.6 | 730 | 2.345 |

### 2.4. Meshing and Numerical Methods

Due to the large scale of the computational physics model, this paper performed a block process of the model when carrying out the meshing in GAMBIT software. Blocks are connected with the interface. It also divides grids independently in each area, using three-dimensional octahedral grids and encrypting wall positions, totaling about 12 million grids. Grid independence analysis and time step verification are shown below. The horizontal view of the meshing results of block grids is shown in Figure 2.

Unsteady computation is performed in the model using commercial software ANSYS FLUENT R15.0. The pressure–velocity coupling adopts the SIMPLE algorithm and the uncoupled implicit solver. The numerical dispersion method controlling the equation

adopts the two-order accuracy upwind difference format. Set "lift" in the "solve-monitor-force". Set the time step as $1 \times 10^{-3}$ s. The convergence residual error is $10^{-4}$.

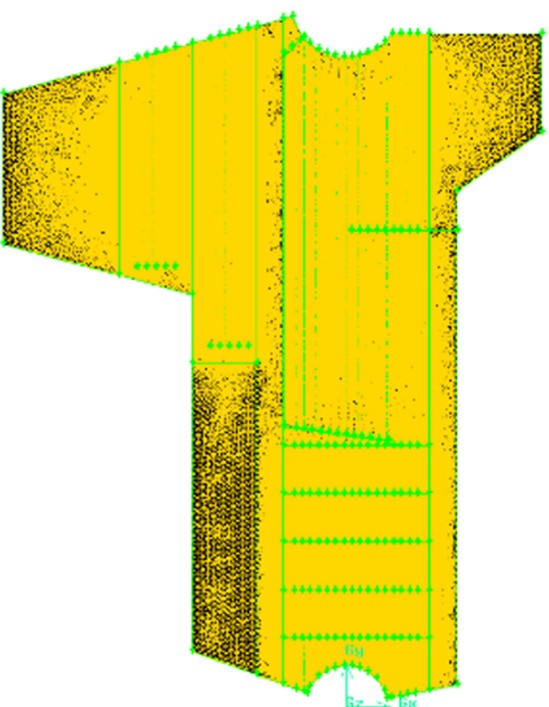

**Figure 2.** Block mesh of the convection bank in a D-type boiler.

In this paper, the grid number and time step selection of the numerical model are verified. According to the influence of different grid numbers and time step changes on frequency under different working conditions, the grid independence analysis and time-step analysis were carried out. Figure 3a shows the selection study of different grid numbers, a total of 6 million, 12 million, and 20 million grids are selected for research. The results show that the error of 6 million grids is about 2.597%, which is relatively large. The 12 million grids selected in this paper have a better fitting effect, so 120,000 mesh is reasonable. The time step is selected as 0.1 s, 0.05 s, and 0.01 s for the study. The research results show that the change of time step has little influence on frequency under different operating loads, as shown in Figure 3b. Therefore, in order to obtain a more accurate numerical solution, the time step in this paper is selected as 0.05 s, which is reliable.

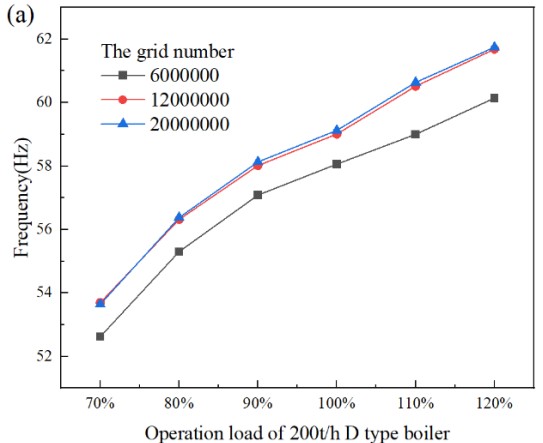
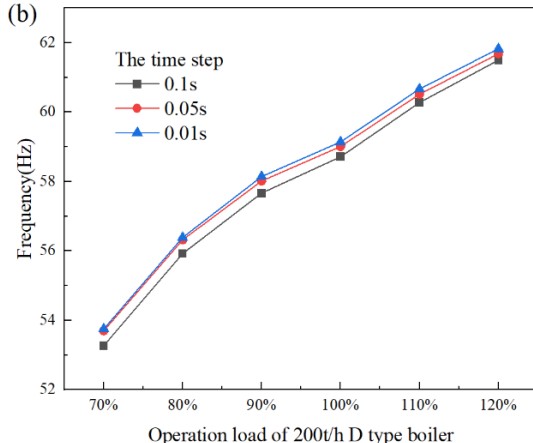

**Figure 3.** Block mesh of the convection bank in a D-type boiler: (**a**) Study on grid Independence; (**b**) Study on time step.

### 3. Results and Discussion

#### 3.1. Numerical Calculation Results

Instead, through the monitoring of the lift coefficient and the fast Fourier transform tool, the lift coefficient time history curves and Karmen vortex shedding curves of the tube bundle's vibration of 200 t/h large-capacity D-type boilers in 120%, 110%, 100%, 90%, 80%, and 70% operation conditions are obtained. The computation results in different operating conditions are shown in Figures 4–9.

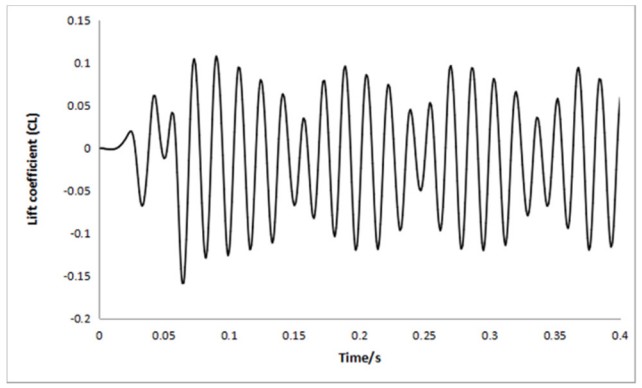
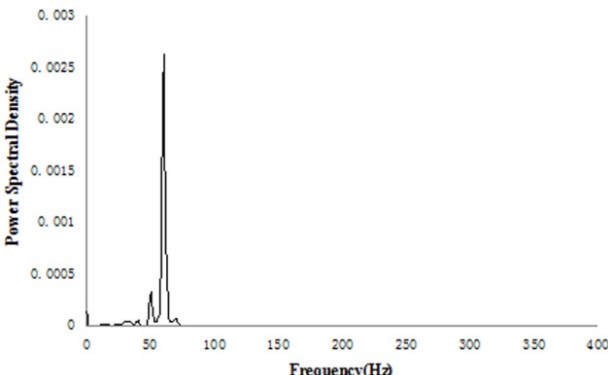

**Figure 4.** The lift coefficient time history curve and power spectrum curve of the vibration area under the 120% operation mode.

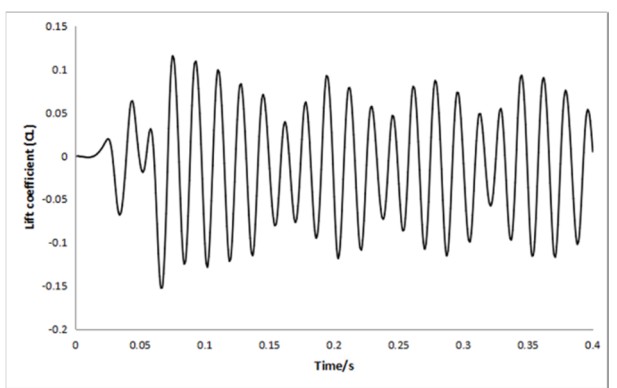
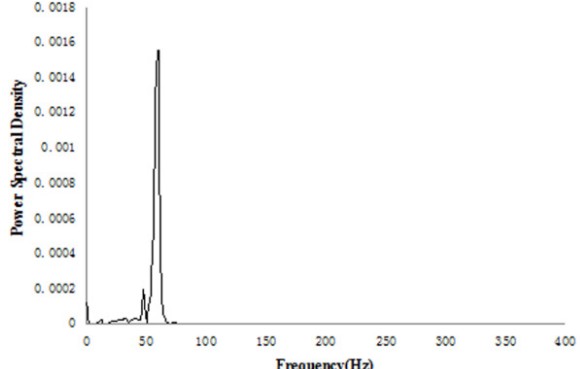

**Figure 5.** The lift coefficient time history curve and power spectrum curve of the vibration area under the 110% operation mode.

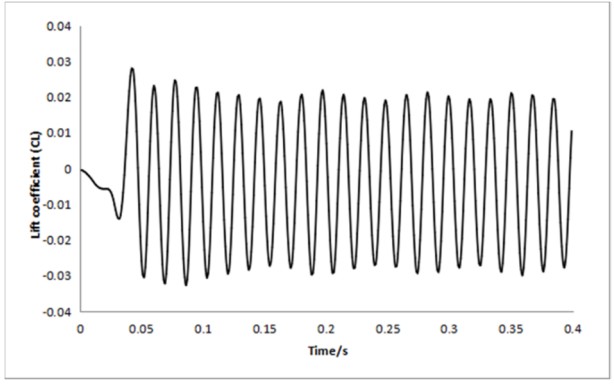
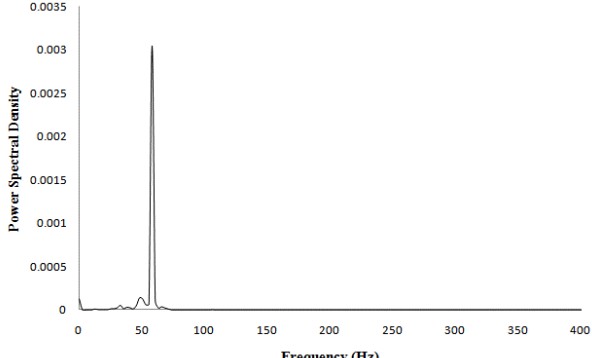

**Figure 6.** The lift coefficient time history curve and power spectrum curve of the vibration area under the 100% operation mode.

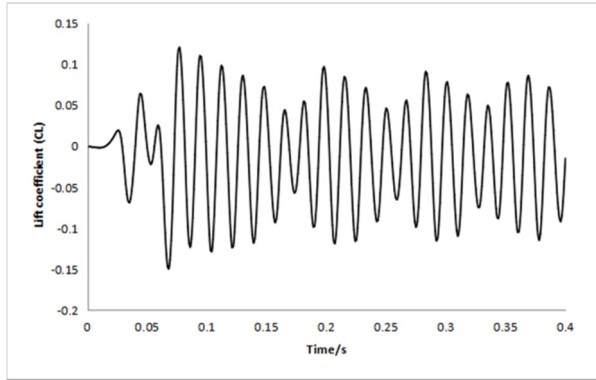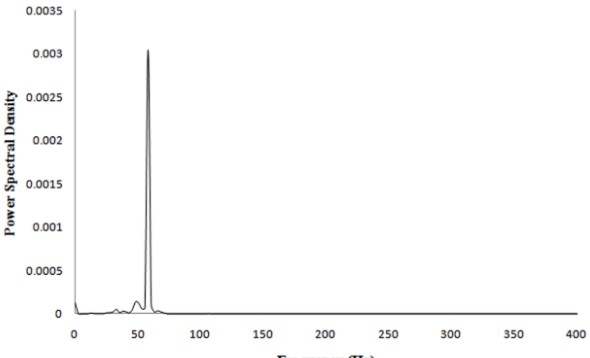

**Figure 7.** The lift coefficient time history curve and power spectrum curve of the vibration area under the 90% operation mode.

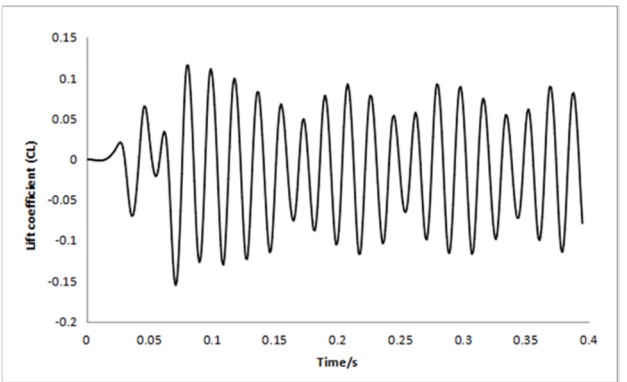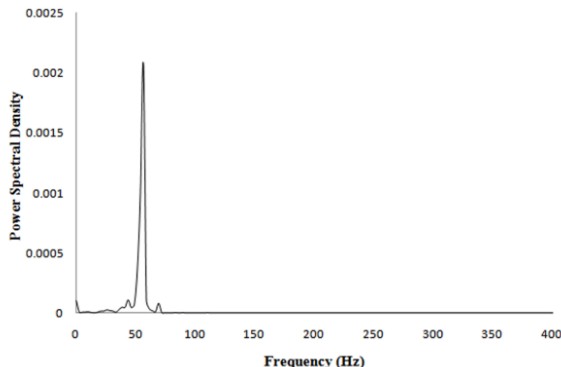

**Figure 8.** The lift coefficient time history curve and power spectrum curve of the vibration area under the 80% operation mode.

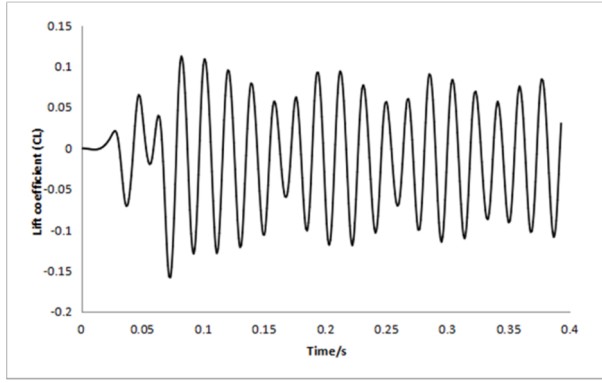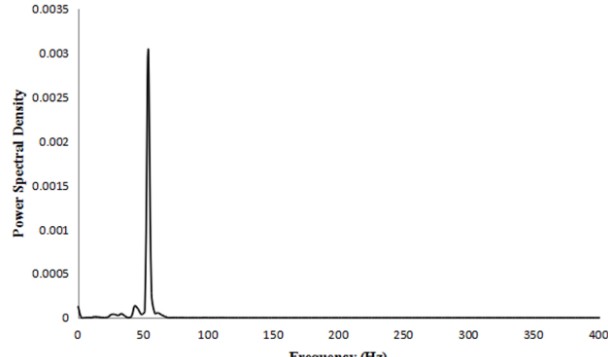

**Figure 9.** The lift coefficient time history curve and power spectrum curve of the vibration area under the 70% operation mode.

The above monitoring results are sorted out, as shown in Table 3.

**Table 3.** Setting of the boundary conditions.

| | Wall Temperature/K | Emissivity $\varepsilon$ (Internal Emissivity) | Wall Roughness/mm |
|---|---|---|---|
| High-temperature superheater | 707 | 0.731 | |
| Low-temperature superheater | 643 | 0.753 | |
| Convection tube bundle | 539 | 0.786 | 0.45 |
| Water wall | 534 | 0.788 | |

According to the following monitoring (Table 4), it can be found that under the 120% working condition, the relatively stable shedding area of the Karman vortex is in the fourth section below the baffle plate under the convection bundle area, and the shedding frequency of the Karman vortex is 61.8 Hz at this time. Under the 110% working condition, the relatively stable shedding area of the Karman vortex is in the fourth section below the baffle plate under the convection bundle area, and the shedding frequency of the Karman vortex is 60.6 Hz at this time. Under the 90% working condition, the shedding area of the relatively stable Karman vortex is in the fourth section below the baffle plate under the convection bundle area, and the shedding frequency of the Karman vortex is 58.1 Hz at this time. Under the 80% working condition, the relatively stable shedding area of the Karman vortex is formed in the third section below the baffle plate under the convection bundle area, and the shedding frequency of the Karman vortex is 56.4 Hz at this time. Under the 70% working condition, the relatively stable shedding area of the Karman vortex is in the third section below the baffle plate under the convection bundle area. At this time, the shedding frequency of the Karman vortex is 53.8 Hz.

**Table 4.** The numerical simulation calculation results of Karmen vortex shedding frequency in different operation conditions.

| Operation Load | 70% | 80% | 90% | 100% | 110% | 120% |
|---|---|---|---|---|---|---|
| Karmen vortex shedding frequency | 53.8 Hz | 56.4 Hz | 58.1 Hz | 59.1 Hz | 60.6 Hz | 61.8 Hz |
| The region that triggers the vibration | Section 3 | Section 3 | Section 4 | Section 4 | Section 4 | Section 4 |

### 3.2. Result Analysis

We compared the Karmen vortex frequency obtained by the numerical simulation in different operating conditions of the 200 t/h D-type boilers with the frequency determined by the traditional Strouhal method and the first-order acoustic standing wave frequency, the result is shown in Figure 10.

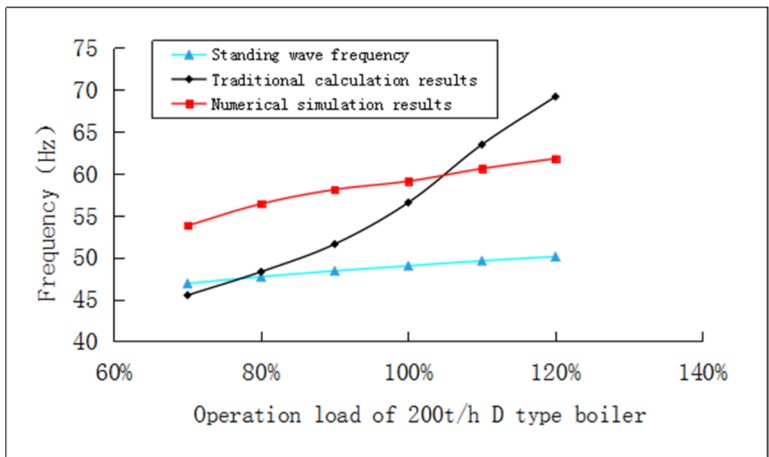

**Figure 10.** Comparison between the first-order acoustic standing wave frequency in different operating conditions and the Karmen vortex shedding frequency obtained with different calculation methods.

It has been pointed out in literature [24] that when the gas flows through the tube bundle's flue, the similarity between the Karmen vortex frequency and the acoustic standing wave frequency is the incentive of vibration. It is considered that when the Karmen vortex shedding frequency is between 0.8 and 1.2 times the standing wave frequency, the tube bundle vibration will be induced.

### 3.2.1. Result Verification

To verify the correctness of the model established in this paper, the comparative experiment about the Karmen vortex shedding frequency and first-order acoustic standing wave frequency with numerical simulation and traditional methods were performed with the 100 t/h D-type gas-fired boilers [25] in the operation of Suzhou Marine & Land Heavy Industry Corp., Ltd. (Zhangjiagang, Jiangsu).

The simulation calculation in Table 4 shows that with the condition declining, the Karmen vortex shedding frequency and the first-order standing wave frequency become closer, which is in accordance with the result that the boiler's vibration becomes more obvious with the operation condition declining. The traditional calculation result of Karmen vortex shedding frequency shows that the largest vibration occurs in the condition of 85–90%; even destructive accidents happen. However, at the site, the obvious vibration phenomenon of the boiler did not occur in the actual operation process within this operation condition interval. Since the traditional calculation results of the Karmen vortex shedding frequency are obtained based on the average gas flow velocity and empirical formulas, the calculation deviation is large. While the numerical simulation results in this paper can determine the position of the Karmen vortex-induced vibration of the boiler's tube bundle area and the range of the coupling vibration more correctly, which is more valuable to guide the design of boiler engineering. Thus, the following prediction analysis will be carried out according to the Karmen vortex shedding frequency obtained by the numerical simulation (Table 5).

**Table 5.** The Karmen vortex frequency's calculation results of 100 t/h D-type boilers in different conditions with different calculation methods.

| Operation Load | 80% | 90% | 100% | 110% | 120% |
|---|---|---|---|---|---|
| Numerical simulation result | 49 Hz | 51 Hz | 52 Hz | 54 Hz | 55 Hz |
| Traditional calculation result | 45 Hz | 47 Hz | 48 Hz | 52 Hz | 54 Hz |
| Vibration area | Section 2 | Section 3 | Section 3 | Section 3 | Section 4 |
| First-order acoustic standing wave frequency | | | 46.35 Hz | | |

### 3.2.2. Analysis and Verdict

From the above numerical simulation and analysis of the Karmen vortex shedding of the convection tube bundle area, in the 100% rated condition, a Karmen vortex with the shedding frequency of 59.1 Hz is formed in Section 4 at the bottom of the lower baffle. Comparing it with the calculated first-order standing wave frequency of 49 Hz, the Karmen vortex shedding frequency is almost the same as slightly over 1.2 times the first-order standing wave frequency. It can be considered that in the rated condition, this 200 t/h boiler will generate an extremely slight resonance phenomenon.

Accordingly, in the 90% (180 t/h), 80% (160 t/h), and 70% (140 t/h) conditions, by calculation, it is discovered that periodic shedding of the Karmen vortex still exists. The Karmen vortex shedding frequency obtained by numerical simulation remains within the range of 0.8 to 1.2 times the first-order standing wave frequency. With the operation load decreasing, the Karmen vortex shedding frequency is closer to the first-order standing wave frequency. Meanwhile, the location where the Karmen vortex shedding is formed moves upward gradually and the Karmen vortex shedding frequency is also decreased. Hence, it can be determined that under the rated load in variable condition operation with the load decreasing, there will be a resonance of the convection tube bundle due to the Karmen vortex shedding. The resonance strength increases with the load condition declining.

By the calculation of the 110% (220 t/h) and the 120% (240 t/h) conditions, periodic shedding of the Karmen vortex still exists. However, the Karmen vortex shedding frequency obtained by numerical simulation is higher than 1.2 times the first-order standing wave frequency. Therefore, it can be determined that over the rated condition, when the load is

increased in operation, there will be no resonance occurring in the convection tube bundle due to Karmen vortex shedding.

In the corresponding 110% (220 t/h) and 120% (240 t/h) working conditions, it is found that periodic shedding Karman vortices still exist, but the shedding frequency of Karman vortices obtained through numerical simulation is higher than 1.2 times the first-order standing wave frequency. Therefore, it can be judged that when the load increases above the rated load in variable operating conditions, the resonance phenomenon of the convection tube bundle does not occur due to the shedding of the Karman vortex.

Compared with the previous calculation results, it can be found that a 100 t/h D-type boiler will cause resonance due to Karman vortex shedding at 80 t/H–120 t/h conditions, and the resonance phenomenon is most significant at 80 t/h condition. In this paper, based on the adjustment of the overall height of the boiler and the structural arrangement of the bundle area, the development of a 200 t/h large-capacity D-type boiler, compared with the original 100 t/h D-type boiler, it can be found that under the rated load of 200 t/h, although the vibration of the boiler is still caused by the shedding of Karman vortex, compared with the original 100 t/h D-type boiler, the vibration is weakened.

### 4. Conclusions

In this paper, numerical simulation analysis is performed on the Karmen vortex shedding-induced vibration of the convection tube bundle area under different conditions of 200 t/h large-capacity D-type boilers. The following conclusions are obtained:

(1) Based on the variable working conditions of a 100 t/h D-type gas-fired boiler, the numerical analysis model established in this paper is reliable. Compared with the Karmen vortex shedding frequency obtained by traditional calculation methods, the Karmen vortex shedding frequency obtained with the numerical simulation method in this paper is closer to the true value of the Karmen vortex shedding frequency.

(2) The simulation results show that with the working condition load decreasing, the Karmen vortex shedding frequency obtained by numerical simulation is decreased and the position where the periodic shedding of the Karmen vortex is formed also moves upward from the bottom area of the lower baffle.

(3) Under the 200 t/h load, at the location of Section 4 at the bottom of the lower baffle, the Karmen vortex shedding frequency is 59.1 Hz. At this time, mild resonance will occur. Taking the 200 t/h load as the boundary, the vibration strength is increased gradually when the operation load is decreased.

**Author Contributions:** S.Y. and L.Z. conceived the main concept. X.H., H.M. and L.Z. contributed to the investigation and data analysis. X.H., L.Z. and X.S. wrote the manuscript. All authors contributed to the writing of the final manuscript. All authors have read and agreed to the published version of the manuscript.

**Funding:** This research received no external funding.

**Institutional Review Board Statement:** Not applicable.

**Informed Consent Statement:** Not applicable.

**Data Availability Statement:** Not applicable.

**Conflicts of Interest:** The authors declare no conflict of interest.

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
