# Peer review of "Analysis and Prediction of Flow-Induced Vibration of Convection Pipe for 200 t/h D Type Gas Boiler"

_axioms, doi:10.3390/axioms11040163_

Round 1
Reviewer 1 Report
The authors focused on an interesting flow problem in the convection pipes of a gas boiler.
The analysis of the problem is focused mainly on mathematical simulations at selected parameters.
I liked the chosen method by which the authors analyzed the vibrations induced by the Carmen vortex.
I have no serious comments on the submitted manuscript. Rather, for visual evaluation, I would recommend the authors to keep the same scales of x and y axes (Fig.3 - Fig.8).
At first glance, big differences are not visible, but if the same scales were maintained, it would be more visible. The evaluation itself is already in the text itself.
A small literature review was also used for the specificity of the problem. However, this does not reduce the level of the submitted manuscript.
Reviewer 2 Report
This manuscript needs some minor improvement before it can be consoidered for publication
--Improve Abstract content and length. Provide more detail.
--Introduction and section 2 would benefit with the inclusion of a figure that details the overall conditions and assumptions
--provide additional design details in Figure 1. Provide additional scale information
--are all variables defined
--provide a sensitivity and error analysis
--Improve English grammar and paper structure (i.e. don't start sentences with the word "And", etc)
--provide mesh size
--discuss grid independence
--discuss operating conditions under the various results (i.e. 120%, 100%, etc)
--Discuss the figures in detail...don't lump all together (i.e Figures 3-8)
--add current (less than 5 years old) references. currently there are 0 of 14 current references. Add additional MDPI references.
--
Round 2
Reviewer 2 Report
This revised manuscript has addressed most of the concerns of this reviewer